# LossAgent: Towards Any Optimization Objectives for Image Processing with LLM Agents

## Abstract

We present the first loss agent, dubbed LossAgent, for low-level image processing tasks, *e.g.*, image super-resolution and restoration, intending to achieve any customized optimization objectives of low-level image processing in different practical applications. Notably, not all optimization objectives, such as complex hand-crafted perceptual metrics, text description, and intricate human feedback, can be instantiated with existing low-level losses, *e.g.*, MSE loss, which presents a crucial challenge in optimizing image processing networks in an end-to-end manner. To eliminate this, our LossAgent introduces the powerful large language model (LLM) as the loss agent, where the rich textual understanding of prior knowledge empowers the loss agent with the potential to understand complex optimization objectives, trajectory, and state feedback from external environments in the optimization process of the low-level image processing networks. In particular, we establish the loss repository by incorporating existing loss functions that support the end-to-end optimization for low-level image processing. Then, we design the optimization-oriented prompt engineering for the loss agent to actively and intelligently decide the compositional weights for each loss in the repository at each optimization interaction, thereby achieving the required optimization trajectory for any customized optimization objectives. Extensive experiments on three typical low-level image processing tasks and multiple optimization objectives have shown the effectiveness and applicability of our proposed LossAgent.

## 1 Introduction

With the revolutionary advancements in deep learning technology, low-level image processing tasks, *e.g.*, image super-resolution and restoration, have garnered increasing interest from researchers. Typically, low-level image processing tasks are optimized with the commonly-used loss function, such as MSE and L1 Losses, in an end-to-end manner, to improve the objective quality (Zamir et al., 2022; Chen et al., 2024; 2023b) or perceptual quality (Yu et al., 2024; Yue et al., 2024; 2025). However, optimizing models using a single optimization objective falls short of meeting real-world needs. For example, in image super-resolution, we desire the super-resolved images to not only restore the ground truth at the pixel level but also to appear natural without artificial textures or visually distracting artifacts (Ledig et al., 2017). To address this, some researchers have introduced the combination of multiple loss functions (Ledig et al., 2017; Wang et al., 2021; Zhang et al., 2021) (*e.g.*, GANs) to train networks, enabling the optimized models to satisfy multiple optimization objectives. Nevertheless, this approach requires the loss functions corresponding to optimization objectives to be differentiable and suitable for training. Consequently, some advanced image quality assessment (IQA) metrics, which align more closely with human visual perception, are not differentiable and thus cannot be directly utilized for end-to-end network optimization.

Recently, large language models (LLMs) such as GPT series (OpenAI, 2023; 2025) and LLaMA series (MetaAI, 2024; Roziere et al., 2023), have shown promising reasoning and understanding capabilities. This has also catalyzed the trend of utilizing LLMs as intelligent agents (Lu et al., 2024; Shinn et al., 2024), especially in the field of embodied AI (Yang et al., 2023a; Gupta & Kembhavi, 2023). By providing the agent with the environment information, predefined settings, rules, external feedback, and a set of optional actions, it can leverage its powerful reasoning capabilities to generate outputs that meet customized requirements, such as tool selection (Shen et al., 2024), action decisions (Yang et al., 2023b), programming (Surís et al., 2023), etc.

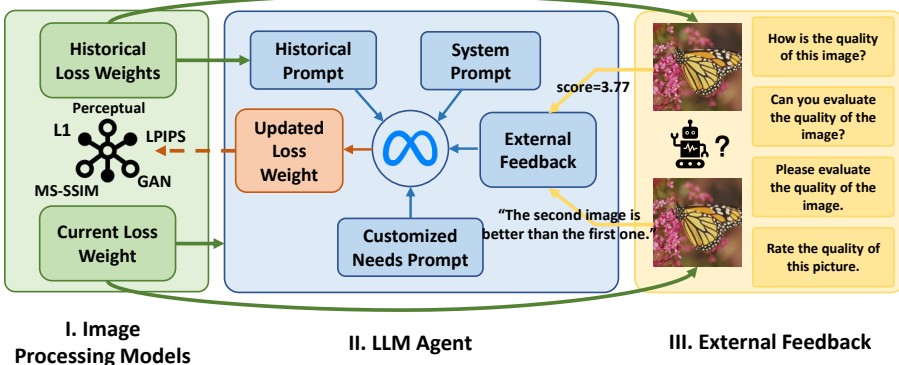

Figure 1: During the training of image processing models (Part I), the loss agent (Part II) gathers feedback from various optimization objectives (Part III). Combining this feedback with historical information, the LLM leverages its powerful reasoning capabilities to determine the optimal loss weights for the subsequent optimization phase of the image processing models (Part I).

Inspired by this series of works, we propose the first loss agent, dubbed LossAgent, for low-level image processing, enabling any customized optimization objectives of the image processing network for multiple practical applications. To achieve this, we introduce the pre-trained large language model (LLM), *i.e.*, LLaMA-3 (MetaAI, 2024) as the loss agent to control the optimization trajectory for different objectives. In the optimization process, an intuitive strategy is to exploit the expected optimization objective as the loss function to guide the optimization of image processing networks. However, not all optimization objectives can assist this, such as the complex hand-crafted optimization objective, textual description, and human feedback, since they cannot be differentiable for end-to-end optimization. To solve the problem, we propose the compositional loss repository, which collects *existing popular loss functions* supported for low-level image processing, and utilize our proposed LossAgent to adaptively and actively assign the weights for each loss at each iteration period based on external environments to achieve customized optimization trajectory toward required optimization objective. In this process, we carefully design the optimization-oriented prompt engineering, which constructs the prompt templates to guide the LLM to understand the current optimization states, trajectory and objectives, thereby achieving accurate loss weights planning. To fully utilize the reasoning capabilities of LLM, the agent receives input of all weights of the model from the beginning of the training phase to the current stage. This enables the LossAgent to smoothly and automatically optimize the image processing model towards predefined optimization objectives through the analysis of historical weights, inference from external feedback, and following customized instructions.

Overall, the LossAgent possesses the following core features:

- LossAgent is capable of obtaining feedback from non-differentiable optimization objectives and leveraging the model's powerful reasoning capabilities to convert this feedback into a composition of loss weights for training, thereby enabling the model to be optimized in an end-to-end manner towards any optimization objectives.

- LossAgent enjoys a high degree of flexibility. Leveraging its powerful reasoning capabilities, the agent can update loss weights fully automatically. Additionally, due to its ability to follow instructions, it can also receive feedbacks from external environments during the training process to pursue customized needs.

- LossAgent exhibits high scalability. As depicted in Figure 1, our LossAgent can be extended to various low-level image processing tasks and multiple different optimization objectives, even if they are not differentiable, which has been proven in the experimental parts.

## 2 RELATED WORKS

### 2.1 IMAGE PROCESSING

Image processing consists of a broad spectrum of tasks, including image restoration (Potlapalli et al., 2023; Guo et al., 2024c), image enhancement (Yu et al., 2024), and image super-resolution (Yue

et al., 2024; Chen et al., 2025). In low-level image processing tasks, pioneering works (Lim et al., 2017; Zhang et al., 2018b) focus primarily on optimizing fidelity-wise metrics, such as PSNR and SSIM, through L1 or MSE loss functions. However, models optimized by these metrics tend to generate over-smoothed results (Ledig et al., 2017). To mitigate this problem, works (Ledig et al., 2017; Wang et al., 2021) leverage generative adversarial networks (GANs) to enable the SR network to learn the distribution of real-world high-quality images. By introducing a weighted combination of VGG perceptual loss (Simonyan & Zisserman, 2014) and GAN loss, GAN-based works (Wang et al., 2018b; Zhang et al., 2021) are well optimized for human perception objectives. More recently, transformer-based (Chen et al., 2023a;b) and diffusion-based works (Wu et al., 2024b) further improve the performance on aforementioned optimization objectives.

However, despite the revolution of network structures and loss function designs, optimization trajectories of image processing models have become relatively fixed. While there is a strong demand for advanced image quality assessment (IQA) metrics (Zhang et al., 2021), many recently developed IQA metrics (You et al., 2024; Wu et al., 2025) cannot be utilized as optimization objectives due to their non-differentiable nature. In this paper, we tackle this challenge by introducing an LLM-based loss agent. This agent is capable of bridging any customized optimization objectives with the combination of loss function weights, allowing for the optimization of image processing models in an end-to-end manner.

## 2.2 LLM Agents

With the development of data science and computing resources, numerous of large language models (LLMs) (Li et al., 2023; Touvron et al., 2023; Brown et al., 2020) have emerged with remarkable language understanding and reasoning abilities. Despite of the above advantages, LLMs may struggle with tasks in certain specialized domains, leading to inaccurate outputs (Ge et al., 2024; Mialon et al., 2023). Consequently, researchers leverage these powerful LLMs as tools planner (Schick et al., 2024) and intelligent agents (Shinn et al., 2024), adaptively coordinating domain-specific expert models based on external demands. For example, MM-REACT (Yang et al., 2023b) tackles various multimodal reasoning and action tasks via prompting ChatGPT (Brown et al., 2020) to invoke domain experts. Schick et al. (2024); Shen et al. (2024) embeds external API tags within text sequences to enhance LLMs' interaction with external resources. With appropriate instruction tuning, researchers have enabled LLMs to adapt to a broader range of tasks, allowing for more specialized task planning (Shen et al., 2024; Surís et al., 2023; Gupta & Kembhavi, 2023; Yang et al., 2023a; Mu et al., 2024). These lines of work demonstrate that the agent is capable of receiving environmental feedback and generating optimal actions accordingly.

Recently, LLMs have garnered attention within the image processing community. For example, LM4LV (Zheng et al., 2024) employs a frozen pre-trained LLM as the backbone for various image restoration tasks. Du et al. (2025) leverages rich prior knowledge from pre-trained LLM to perform lossless image compression.

Different from these great efforts, we propose the first LLM-based agent to handle any customized optimization objectives for image processing models, named LossAgent. By leveraging the powerful understanding and reasoning capabilities of LLMs, we transform feedback from external models or metrics into appropriate adjustments of loss weights for image processing models, allowing image processing models to be *optimized* toward any objectives. We hope that our LossAgent will facilitate the development of image processing to be more open-ended and intelligent.

## 3 Methods

### 3.1 Motivation

Although the network structures of image processing models have evolved significantly in recent years, the optimization objectives of these models have remained largely unchanged. Taking image super-resolution (ISR) as an example, early works (Lim et al., 2017; Dong et al., 2015; Zhang et al., 2018b) pursued higher PSNR values, while some recent works (Yue et al., 2024; Wu et al., 2024b; Chen et al., 2025) have started optimizing networks to better align with human perception considering metrics such as LPIPS (Zhang et al., 2018a) and NIQE (Mittal et al., 2012). Despite advances in these ISR models, image quality assessment (IQA) models have concurrently experienced significant

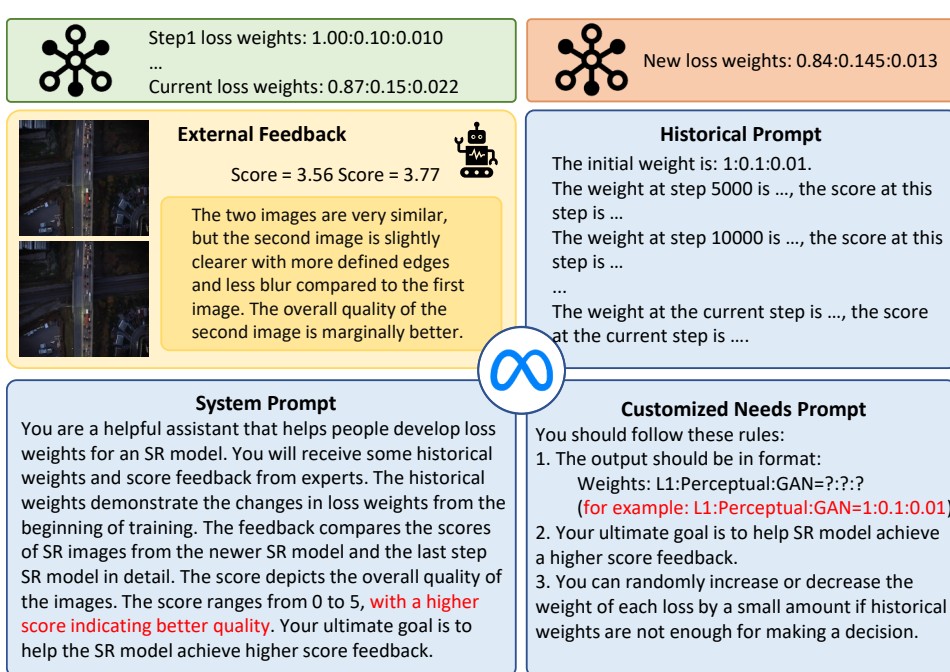

Figure 2: The overview of LossAgent. LossAgent bridges image processing models with any optimization objectives through the following workflow: The image processing model will generate images using weights at the current stage. Subsequently, external expert model will generate scores or textual feedback according to the images provided by the image processing model. The LLM-based agent model (*e.g.*, LLaMA3) collects feedback and leverages its powerful reasoning abilities to analyze the relationships between loss weights and optimization objectives while following our prompt engineering, including system prompt, historical prompt, and customized needs prompt. After proper analysis, the agent will generate a new combination of loss weights to further guide the next step in optimizing the image processing model. We provide a detailed **case study** in Appendix A.4.

developments. An IQA model evaluates the visual quality of images by analyzing their attributes and detecting any distortions or imperfections, making it particularly suitable as an optimization objective for image processing models (Wang et al., 2023; Yang et al., 2022). Given the general unavailability of reference images in practical applications, most state-of-the-art IQA metrics adopt the no-reference (NR) paradigm.

However, directly applying NR metrics as loss function for image processing model results in unstable training and suboptimal results, since they fail to effectively guide the model in capturing the structural information of images. Nevertheless, due to the specific operations in large model-based IQA metrics (*e.g.*, incorporating other models and applying sampling (Wu et al., 2024a; You et al., 2024; Wu et al., 2025)), some advanced IQA metrics are non-differentiable, preventing them from being utilized as the optimization objectives during the training of image processing models. Moreover, when leveraging textual feedback from humans or MLLM-based IQA models such as Co-Instruct (Wu et al., 2024a) for optimization objectives, the metrics derived from these objectives are inherently non-differentiable. Consequently, we raise a significant and interesting question "How to optimize an image processing model with these advanced IQA metrics?"

In this paper, we address the above challenges by introducing an LLM-based agent, as shown in Figure 1. Instead of directly applying these optimization objectives as loss functions for training image processing models, LossAgent efficiently transfers various forms of feedback from customized optimization objectives into an actionable weighted composition of commonly used loss functions.

## 3.2 WEIGHTED COMPOSITIONAL LOSS REPOSITORY

To achieve any optimization trajectory in the training stage of image processing models, we establish the compositional loss repository with multiple typical differential loss functions

$\{L_1, L_2, L_3, ..., L_M\}$, such as $L_1$ and LPIPS losses, where the weighted composition of them with coefficients $\{w_1, w_2, w_3, ..., w_M\}$ is formed to the final loss for optimizing image processing models:

$$\mathcal{L} = w_1 L_1 + w_2 L_2 + \cdots + w_M L_M. \tag{1}$$

Here, $M$ is the total number of loss functions. Based on the above weighted compositional loss repository, we can adjust the optimization direction directly by generating the weighting coefficients through our proposed loss agent. To enable the loss agent to adjust weight composition in time based on feedback from any optimization objective, we divide the training stage of the image processing model into $N$ stages, where the current state of the image processing model and their corresponding compositional loss is as:

$$\mathcal{S} = \{S_0, S_1, S_2, \ldots, S_i, \ldots, S_N\}, \tag{2}$$

$$\mathcal{L}_i = w_1^i L_1 + w_2^i L_2 + \cdots + w_M^i L_M, \tag{3}$$

where $S_0$ stands for the initial states of the image processing model and $i$ indicates the $i^{th}$ training stage. The external feedback will be generated by the optimization objectives at the end of each training stage with a set of randomly selected testing images as:

$$\mathcal{I} = \{I_1, I_2, \ldots, I_T\}, \tag{4}$$

where $T$ is the number of images. We provide the details for $\mathcal{I}$ in the **Datasets** part of Section 4.1.

### 3.3 External Feedback

To alleviate the cognitive burden on the loss agent for the image processing task, we introduce the external evaluation expert $\mathcal{O}$ to produce the optimization feedback to the loss agent. Concretely, once we obtained the restored images $\mathcal{I}_{S_i}$ at the stage $S_i$, we can utilize external evaluation expert $\mathcal{O}$ to evaluate the quality of restored images $\mathcal{I}_{S_i}$ as:

$$\mathcal{F} = \mathcal{O}\left(\mathcal{I}_{S_i}\right), \tag{5}$$

where $\mathcal{F}$ is the external feedback from optimization objectives, which can be a quality score or textual description. Notably, the external evaluation expert is the tool to represent the optimization objective. For instance, if the optimization objective is to achieve a higher CLIPIQA (Wang et al., 2023) score, we select CLIPIQA as the external evaluation expert. Conversely, when the optimization objective is more general (*e.g.*, to achieve higher quality), multiple evaluation experts can be utilized collaboratively to generate feedback. We provide more details in Section 4.2.2.

### 3.4 Loss Agent

It is noteworthy that the current LLM cannot be directly applied to image processing tasks due to the knowledge discrepancy. To equip the LLM with the capability to understand the image processing task and adjust the optimization direction of image processing, we further exploit prompt engineering to adapt the pre-trained LLM to our desired loss agent. Concretely, our proposed prompt engineering strategy can be divided into three parts: i) **system prompt**, ii) **historical prompt**, and iii) **customized needs prompt**.

After feedback $\mathcal{F}$ is generated from external expert models, the loss agent will collect and utilize this feedback to update a new set of loss weights. LLM demonstrates exceptional capabilities in following instructions and making decisions (Shen et al., 2024; OpenAI, 2023; Touvron et al., 2023). Consequently, enabling the loss agent to accomplish our task is feasible by providing accurate and sufficient prompt guidance. Initially, we employ prompt engineering through **system prompt** approach following previous works (Shen et al., 2024; Yang et al., 2023a; Mu et al., 2024; Surís et al., 2023) to convey to the loss agent the role it needs to undertake, the inputs it will receive, the required outputs, and the objectives to be achieved. An example of our prompt engineering under the ISR scenario is given in Figure 2. The most important instruction for the agent is the objectives clarification: "*Your ultimate goal is to help the SR model achieve higher score feedback.*". This is because LLM may not encompass the knowledge of how these IQA metrics should be evaluated. Therefore, it is crucial to clarify whether lower or higher scores indicate better image quality. Without this context, LLM might intuitively assume that higher scores indicate better quality, resulting in incorrect reasoning (Table 5).

Table 1: Quantitative comparisons between LossAgent and other methods on classical image SR. "Pre-trained" denotes the pre-trained weight. "Random" denotes that we randomly update loss weights. "Fixed" denotes that we train the model with fixed loss weights. As NIQE (Mittal et al., 2012), MANIQA (Yang et al., 2022), CLIPIQA (Wang et al., 2023), and Q-Align (Wu et al., 2023) are no-reference IQA metrics, we also calculate these metrics for ground-truth (GT) as a reference. ↑ / ↓ indicates higher/lower is better. The best results are **bolded**.

| Metrics | Methods | Datasets | | | | | Avg. |
|---------|---------|------|-------|--------|---------|----------|------|
| | | Set5 | Set14 | BSD100 | Urban100 | Manga109 | |
| NIQE↓ | Pre-trained | 7.10 | 6.22 | 6.11 | 5.46 | 5.37 | 6.05 |
| | Random | 5.12 | 4.16 | 4.07 | 4.08 | 3.99 | 4.28 |
| | Fixed | 5.09 | 4.07 | 3.99 | 4.04 | 3.95 | 4.23 |
| | LossAgent | **4.82** | **3.91** | **3.86** | **3.96** | **3.88** | **4.08** |
| | GT (Ref.) | 5.15 | 4.86 | 3.19 | 4.02 | 3.53 | 4.15 |
| MANIQA↑ | Pre-trained | 0.446 | 0.409 | 0.349 | 0.482 | **0.446** | 0.426 |
| | Random | 0.437 | 0.391 | 0.334 | 0.470 | 0.389 | 0.404 |
| | Fixed | 0.458 | 0.406 | 0.354 | 0.494 | 0.416 | 0.425 |
| | LossAgent | **0.474** | **0.418** | **0.365** | **0.496** | 0.424 | **0.436** |
| | GT (Ref.) | 0.534 | 0.449 | 0.523 | 0.552 | 0.420 | 0.496 |
| CLIPIQA↑ | Pre-trained | 0.605 | 0.517 | 0.534 | 0.501 | 0.637 | 0.559 |
| | Random | 0.738 | 0.663 | 0.584 | 0.583 | 0.674 | 0.648 |
| | Fixed | 0.765 | 0.694 | 0.649 | 0.624 | 0.710 | 0.688 |
| | LossAgent | **0.788** | **0.718** | **0.679** | **0.643** | **0.729** | **0.711** |
| | GT (Ref.) | 0.807 | 0.740 | 0.756 | 0.675 | 0.700 | 0.736 |
| Q-Align↑ | Pre-trained | 3.03 | 3.29 | 2.98 | 4.38 | 3.65 | 3.47 |
| | Random | 2.99 | 3.32 | 3.15 | 4.40 | 3.65 | 3.50 |
| | Fixed | 3.04 | 3.45 | 3.34 | **4.53** | **3.66** | 3.60 |
| | LossAgent | **3.07** | **3.48** | **3.41** | **4.53** | 3.65 | **3.63** |
| | GT (Ref.) | 3.36 | 3.63 | 4.04 | 4.53 | 3.60 | 3.83 |

Subsequently, to mitigate the hallucination phenomenon in LLM and prevent undesirable responses in situations of information scarcity, we gather the optimization trajectory of the loss agent as **historical prompt** and provide this information as context to the LLM.

Following this, we impose certain *rule-based constraints* on LLM through **customized needs prompt**. Furthermore, we incorporate format regularization into these rules to alleviate the challenge of parsing LLM outputs, which we found to be highly effective in standardizing the outputs. It is noteworthy that the design of such **customized needs prompt** not only provides flexibility for current usage but also accommodates a variety of future needs.

Ultimately, the loss agent consolidates all received information, leveraging its robust understanding and reasoning capabilities to generate a new set of loss weights as:

$$\mathcal{L}_{i+1} = w_1^{i+1} L_1 + w_2^{i+1} L_2 + \cdots + w_M^{i+1} L_M. \tag{6}$$

This new combination of loss functions will be employed to optimize the image processing model at stage $i + 1$. Based on the system prompt, the historical prompt, and the customized needs prompt, our LossAgent is capable of *updating reasonable new loss weights* for training image processing model. Please refer to Section 4.3 for more details.

## 4 EXPERIMENTS

### 4.1 SETTINGS

To demonstrate the effectiveness of our LossAgent, we perform the evaluation on three representative low-level image processing tasks: classical image super-resolution, real-world image super-resolution, and all-in-one image restoration. We adopt two typical image processing models: SwinIR (Liang et al., 2021) for super-resolution tasks and PromptIR (Potlapalli et al., 2023) for all-in-one restoration task. Notably, more results on various CISR backbones are provided in Appendix A.3.6. To demonstrate the effectiveness of LossAgent towards various optimization objectives, we assess the performance of our method across two testing settings: single optimization objective and double optimization objectives. For all score-based IQA optimization objectives, we adopt their `pyiqa` python implementation (Chen

Table 2: Details of training iterations for each stage, total number of training iterations, and initial weights of loss functions for three image processing models. **For "Fixed" methods, the initial weights are fixed during the training process**.

| Task | Iters. for Each Stage | Total Iters. | Initial Loss Weights |
|---|---|---|---|
| CISR | 5000 | 100k | $\mathcal{L} = 1.0 L_{\text{L1}} + 0.1 L_{\text{per}} + 0.01 L_{\text{GAN}}$ |
| RISR | 5000 | 200k | $\mathcal{L} = 1.0 L_{\text{L1}} + 0.1 L_{\text{per}} + 0.01 L_{\text{GAN}}$ |
| AIR | 2500 | 100k | $\mathcal{L} = 1.0 L_{\text{L1}} + 0.1 L_{\text{per}} + 1.0 L_{\text{LPIPS}}$ |

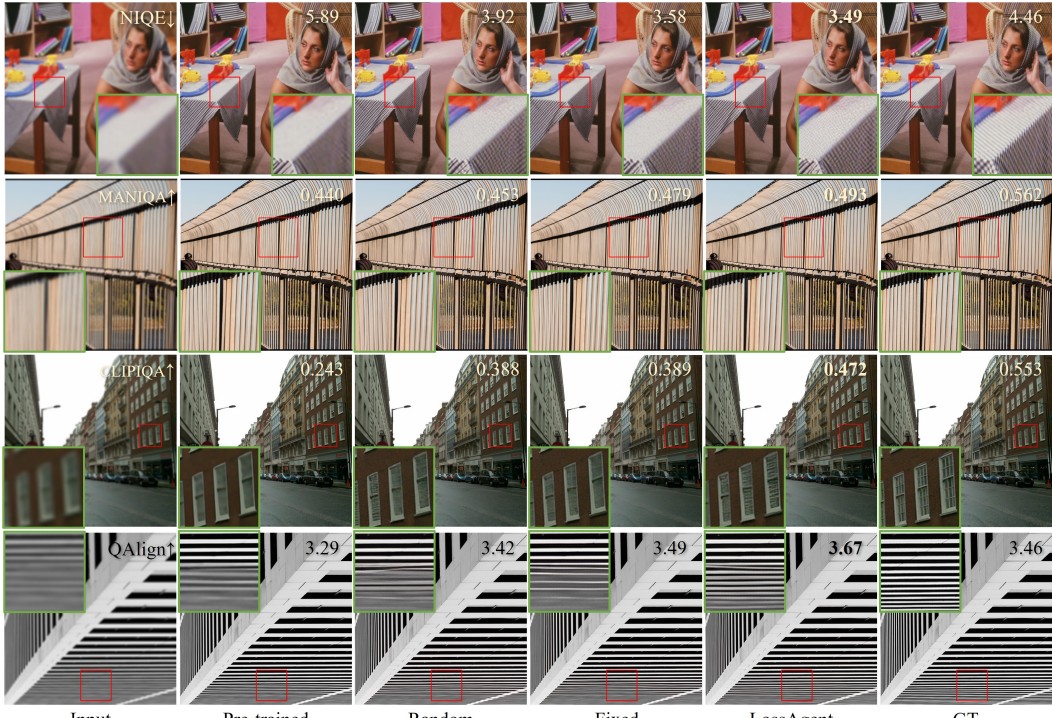

Figure 3: Qualitative comparisons between other methods and LossAgent on CISR. Zoom in for better views.

& Mo, 2022). We select open-sourced `Meta-Llama-3-8B-Instruct`[1] as the LLM of our loss agent due to its impressive reasoning capabilities. We provide the training details in Appendix A.1. To simplify the reasoning process of LLM, we set the number of loss functions in the loss repository to $M = 3$ in all experiments. We mainly compare our LossAgent with three methods: i) "Pre-trained": the official weights provided by each image processing model. ii) "Random": the loss weights are randomly updated in each stage. iii) "Fixed": the loss weights are fixed during the training process, the fixed values are demonstrated in Table 2.

**Datasets.** For image SR tasks, we follow previous works (Liang et al., 2021; Wang et al., 2021) and adopt DF2K (Agustsson & Timofte, 2017; Timofte et al., 2017) as the training dataset. For all-in-one image restoration task, we follow (Li et al., 2022; Potlapalli et al., 2023) to use a combination of BSD400 (Arbelaez et al., 2010), WED (Ma et al., 2016), Rain100L (Yang et al., 2020) and SOTS (Li et al., 2018) to optimize the model. We utilize five SR benchmarks with ground-truth to evaluate the performance of LossAgent on classical image SR: Set5 (Bevilacqua et al., 2012), Set14 (Zeyde et al., 2010), BSD100 (Martin et al., 2001), Urban100 (Huang et al., 2015) and Manga109 (Matsui et al., 2017). Two real-world benchmarks without ground-truth are adopted to evaluate real-world image SR: OST300 (Wang et al., 2018a) and RealSRSet (Zhang et al., 2021). We follow PromptIR (Potlapalli et al., 2023) to use SOTS(test) (Li et al., 2018), Rain100L(test) (Yang et al., 2020) and BSD68 (Martin et al., 2001) to evaluate the all-in-one image restoration performance. For testing images $\mathcal{I}$ mentioned in Equation 4, we randomly sample 10 images from Set14 (Zeyde et al., 2010) for classical image SR;

[1]https://huggingface.co/meta-llama/Meta-Llama-3-8B-Instruct

Table 3: Quantitative comparisons between LossAgent and other methods on real-world image SR. The best results are **bolded**. Notice that, there is no ground-truth for this task.

| Methods | Metrics | Datasets | | Avg. | Metrics | Datasets | | Avg. |
|---|---|---|---|---|---|---|---|---|
| | | OST300 | RealSRSet | | | OST300 | RealSRSet | |
| Pre-trained | | 6.31 | 7.62 | 6.96 | | 0.332 | 0.360 | 0.346 |
| Random | NIQE↓ | 4.17 | 5.56 | 4.86 | MANIQA | 0.341 | 0.366 | 0.353 |
| Fixed | | 3.26 | 5.12 | 4.19 | | 0.366 | 0.385 | 0.375 |
| LossAgent | | **3.05** | **4.43** | **3.74** | | **0.371** | **0.394** | **0.383** |
| Pre-trained | | 4.47 | 3.43 | 3.95 | | 0.419 | 0.444 | 0.432 |
| Random | Q-Align | 4.39 | 3.73 | 4.06 | CLIPIQA | 0.357 | 0.407 | 0.382 |
| Fixed | | 4.55 | 3.81 | 4.18 | | 0.528 | 0.611 | 0.569 |
| LossAgent | | **4.58** | **3.87** | **4.22** | | **0.571** | **0.649** | **0.610** |

randomly sample 10 images from RealSRSet (Zhang et al., 2021) for real-world image SR; randomly sample 10 images from evaluation sets of PromptIR for all-in-one IR.

## 4.2 EVALUATION ON OPTIMIZATION OBJECTIVES

### 4.2.1 SINGLE OPTIMIZATION OBJECTIVE

In this section, we validate the effectiveness of LossAgent towards the single optimization objective. We select four IQA metrics as the optimization objective: NIQE (Mittal et al., 2012), MANIQA (Yang et al., 2022), CLIPIQA (Wang et al., 2023), and Q-Align (Wu et al., 2023). For each metric, to simplify the reasoning of LLM, we start from the pre-trained weights with initial loss weights listed in Table 2, and optimize the image processing model using LossAgent with external feedback from each metric. As demonstrated in Table 1 and 3, our LossAgent outperforms "Random" and "Fixed" method (*i.e.*, fixed loss weights) across almost all the benchmarks under all the optimization objectives, which not only reveals the effectiveness of LossAgent but also indicates that our method enjoys plausible generalization abilities across different image processing models (Due to limited space, we provide results of All-in-one image restoration in Appendix A.2). Notably, LossAgent performs well on real-world image SR task, suggesting the efficacy of our proposed method in complex application scenarios. Specifically, LossAgent outperforms the "Random" method, demonstrating that the proposed LossAgent does not update loss weights arbitrarily. On the contrary, LossAgent effectively leverages the provided prompt information to make reasonable adjustments on loss weights.

We provide qualitative comparisons between other methods and our LossAgent on classical image super-resolution task in Figure 3. As observed, image processing model restores images that more aligned with human perception with the help of LossAgent, while achieving best IQA scores across various metrics.

Table 4: Quantitative comparisons between single and double optimization objectives. For the latter situation, we include both Q-Align score and PSNR value as external feedback for LossAgent.

| Methods | Datasets | | | | | Avg. |
|---|---|---|---|---|---|---|
| | Set5 | Set14 | BSD. | Urban. | Manga. | |
| Q-Align↑ | 3.07/30.62 | 3.48/27.28 | 3.41/26.41 | 4.53/25.96 | 3.65/29.91 | 3.63/28.04 |
| Q-Align↑+PSNR↑ | 3.12/31.14 | 3.46/27.52 | 3.42/26.62 | 4.53/26.27 | 3.65/30.29 | 3.64/28.37 |

### 4.2.2 DOUBLE OPTIMIZATION OBJECTIVES

To fully explore the potential of LossAgent, we conduct an experiment on classical image SR task. In this experiment, we utilize two optimization objectives (*i.e.*, Q-Align (Wu et al., 2023) and PSNR) simultaneously to adjust loss weights. Notably, striking a balance between objective metric (*i.e.*, PSNR) and subjective metric (*i.e.*, Q-Align) is not intuitive since these two metrics are not positively correlated. However, as observed from Table 4, including PSNR as an optimization objective yields PSNR gains across all benchmarks while maintaining comparable or better Q-Align performance. We attribute this to the powerful reasoning capabilities of LLM. Such results showcase the flexibility of LossAgent towards multiple optimization objectives.

**Summary.** We have validated the flexibility and scalability of LossAgent in this Section through two evaluation settings: single optimization objective and double optimization objectives. As

Table 5: Effectiveness of **system prompt**. "W/o" represents that we remove descriptions about the relationship between scores and the qualities of images from the system prompt. Evaluating on CISR task with NIQE↓.

| Sys. Prompt | Datasets | | | | | Avg. |
| --- | --- | --- | --- | --- | --- | --- |
| | Set5 | Set14 | BSD. | Urban. | Manga. | |
| w/o | 5.12 | 4.24 | 4.02 | 4.17 | 4.06 | 4.32 |
| w/ | 4.82 | 3.91 | 3.86 | 3.96 | 3.88 | 4.08 |

Table 6: Effectiveness of **historical prompt**. $S_i$ represents the current stage, while $S_0$ represents the initial stage. We perform evaluation with MANIQA metric.

| Trajectories | Datasets | | | Avg. |
| --- | --- | --- | --- | --- |
| | Set14 | Urban. | Manga. | |
| $\{S_{i-1}, S_i\}$ | 0.405 | 0.487 | 0.413 | 0.435 |
| $\{S_0, \ldots, S_i\}$ | 0.418 | 0.496 | 0.424 | 0.446 |

observed, our LossAgent is efficient towards multiple image processing tasks and various optimization objectives, which also bridges advanced IQA metrics with image processing models. Due to limited space, we provide more ablation studies about LossAgent in Appendix A.3.

### 4.3 EVALUATION ON EFFECTIVENESS OF PROMPT DESIGN

As described in Section 3.4, we carefully devise prompts for the LLM to prevent hallucination and generate reasonable loss weights. Our prompt design mainly focuses on three parts: i) **System prompt** clarifies the roles and goals of LLM. Most importantly, it provides a brief introduction to these IQA metrics about whether lower or higher scores indicate better image quality. ii) **Historical prompt** accommodates previous optimization trajectories, furnishing rich context for the LLM to infer reasonable loss weights. iii) **Customized needs prompt** gives rule-based constraints on LLM's reasoning process.

**Effectiveness of System Prompt.** In Table 5, we remove the prompt that describes the relationship between scores and the qualities of images. Take NIQE (Mittal et al., 2012) as an example, where a lower score indicates a better quality, LossAgent fails to improve the performance of the ISR model on the NIQE metric. We attribute this to the LLM potentially interpreting a higher score as an indicator of better quality. Consequently, our system prompt design helps mitigate hallucination in the decision-making process of LossAgent.

**Effectiveness of Historical Prompt.** Although LLM possesses strong reasoning and decision-making capabilities, it is unable to generate rational loss weights effectively without sufficient context. Therefore, we provide such context by collecting all historical optimization trajectories. As demonstrated in Table 6, providing full historical information through prompt achieves the best performance, while providing only two trajectories (*i.e.*, loss weights and feedback at stage $S_i$ and $S_{i-1}$) leading to performance drops.

**Effectiveness of Customized Needs Prompt.** As LLM generates textual outputs, it is necessary to standardize its outputs by rule-based constraints, making the weights identifiable by programs. We empirically find that given an example of the format (as shown in Figure 2) effectively reduces hallucination in LLM's outputs. Removing this example leads to a significant drop in the successful rate of generating standardized output. In contrast, our LossAgent successfully generates standardized output, with only one failure case out of 800 testing samples. This demonstrates the effectiveness of our customized needs prompt design.

**Summary.** In this Section, we have validated the effectiveness and significance of our prompt design. As demonstrated through ablations of system prompt, historical prompt and customized needs prompt, our LossAgent is capable of surpassing hallucinations and generating reasonable loss weights for image processing models.

## 5 CONCLUSION

In this paper, we propose the first loss agent to address any customized optimization objectives for low-level image processing tasks. By introducing powerful LLM as the loss agent, our LossAgent is capable of understanding various optimization objectives, trajectories, and stage feedback from external expert models. To make the most of the reasoning abilities of LLM, we carefully design the optimization-oriented prompt engineering for the LossAgent by providing detailed instructions along with historical information to prevent hallucinations and incorrect reasoning caused by the LLM. Extensive experiments on three representative low-level image processing tasks with various customized optimization objectives have demonstrated the flexibility and scalability of LossAgent.

ETHICS STATEMENT

Our research on LossAgent is purely algorithmic and all experiments were conducted on publicly available benchmark datasets. This study did not involve human or animal subjects, and we foresee no direct negative societal impacts or ethical concerns arising from our work.

REPRODUCIBILITY STATEMENT

To ensure reproducibility, all source code for our method, along with the experimental scripts, will be made available in a public repository upon publication. The datasets used for experiments are standard public benchmarks, and our repository will include detailed instructions to replicate all reported results.

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

# A  APPENDIX

## A.1  TRAINING DETAILS

As demonstrated in Section 3.2, we divide the whole training process of image processing models into several stages to enable the dynamic adjustment of loss weights through LossAgent. We list the details of training iterations for each stage, the total number of training iterations, and the initial weights of loss functions in Table 1 of the main paper. For two image super-resolution tasks, we utilize the PSNR-oriented pre-trained checkpoints of SwinIR (Liang et al., 2021) as initial checkpoints for both tasks and then apply popular losses from GAN-based training strategies for image SR tasks using our LossAgent. For all-in-one image restoration tasks, we adopt the pre-trained checkpoint of PromptIR (Potlapalli et al., 2023) as the initial checkpoint. However, since GAN-based training is uncommon for this task, we use a combination of L1 loss, perceptual loss, and LPIPS loss as the loss repository to evaluate the performance of our LossAgent. The rationale behind utilizing pre-trained checkpoints as initial checkpoints is to mitigate unstable fluctuations in the early stages of training of image processing models. Such fluctuations may otherwise misguide the LossAgent due to the limited capabilities of current LLMs, leading to inaccurate updates of loss weights. It is noteworthy that, to avoid the effect from the learning rate of the optimizer on our experiments, we uniformly set the learning rate to 1e-4 for all three tasks and kept it constant throughout the training process. Following previous implementations, we utilize an Adam optimizer for each task. We use 8 NVIDIA TESLA V100 GPUs for our experiments, with a total batchsize of 32 for image SR tasks and a total batchsize of 16 for all-in-one image restoration tasks.

## A.2  ADDITIONAL RESULTS FOR ALL-IN-ONE IMAGE RESTORATION

We provide the quantitative results of LossAgent compared to other methods on all-in-one image restoration tasks across four optimization objectives in Table 7. As demonstrated in the Table, in the all-in-one IR task, LossAgent does not perform as robustly as in the other two tasks. We attribute this to the minimal differences between images generated in consecutive stages, which limit the instructional information available to the agent from feedback and hinder its ability to conduct thorough analysis and inference to adjust loss weights. However, LossAgent still shows commendable performance improvement against other methods, indicating the flexibility and adaptability of our LossAgent across different image process models.

## A.3  MORE ABLATION STUDIES

In this section, we provide more ablation studies to verify the reliability of our design for LossAgent.

### A.3.1  ITERATIONS FOR EACH STAGE

In this part, we conduct ablation studies about training iterations for each stage. As demonstrated in Table 8, a moderate choice of 5000 training iterations for each stage achieves the best results. If the iterations are small (i.e., 2500), when reaching the end of training, the list of historical loss weights tends to become very long, thus making it difficult to perform reasoning. If the iterations are large (i.e., 10000), the total update steps tend to be insufficient for a reasonable adjustment of loss weights during training, thereby causing suboptimal results. Therefore, we select the optimal iteration steps for the classical image SR task to be 5000. We apply the same principle to the other two tasks, as listed in Table 1 of the main paper.

Table 7: Quantitative comparisons between LossAgent and other methods on all-in-one IR. The best results are **bolded**.

| Metrics | Methods | Dehaze | Derain | Denoise | | | Avg. |
| | | SOTS | Rain100L | $\sigma = 15$ | $\sigma = 25$ | $\sigma = 50$ | |
|---|---|---|---|---|---|---|---|
| NIQE↓ | Pre-trained | **2.91** | **3.16** | 3.77 | 3.96 | 4.25 | 3.61 |
| | Fixed | 2.98 | 3.18 | 3.43 | 3.61 | 3.88 | 3.42 |
| | LossAgent | 2.95 | 3.17 | **3.38** | **3.48** | **3.80** | **3.36** |
| | GT (Ref.) | 2.94 | 3.17 | 3.13 | 3.13 | 3.13 | 3.10 |
| MANIQA↑ | Pre-trained | 0.441 | 0.498 | **0.493** | 0.457 | 0.377 | 0.453 |
| | Fixed | 0.447 | 0.503 | 0.482 | 0.450 | 0.381 | 0.453 |
| | LossAgent | **0.450** | **0.505** | 0.491 | **0.462** | **0.386** | **0.459** |
| | GT (Ref.) | 0.442 | 0.509 | 0.525 | 0.525 | 0.525 | 0.505 |
| CLIPIQA↑ | Pre-trained | 0.494 | 0.750 | 0.686 | 0.672 | 0.640 | 0.649 |
| | Fixed | 0.534 | 0.769 | 0.795 | **0.785** | **0.725** | **0.722** |
| | LossAgent | **0.542** | **0.771** | **0.807** | 0.777 | 0.706 | 0.721 |
| | GT (Ref.) | 0.544 | 0.755 | 0.757 | 0.757 | 0.757 | 0.714 |
| Q-Align↑ | Pre-trained | 4.02 | 3.92 | **4.09** | 3.96 | 3.61 | 3.92 |
| | Fixed | **4.03** | 3.94 | 3.95 | 3.94 | 3.76 | 3.92 |
| | LossAgent | 3.99 | **3.95** | 3.97 | **3.96** | **3.82** | **3.94** |
| | GT (Ref.) | 3.96 | 4.01 | 4.11 | 4.11 | 4.11 | 4.08 |

Table 8: Quantitative comparisons between different iterations for each stage. Results are reported on classical image SR (CISR) using Q-Align score. The best results are **bolded**.

| Iters. | Datasets | | | | | Avg. |
| | Set5 | Set14 | BSD100 | Urban100 | Manga109 | |
|---|---|---|---|---|---|---|
| 2500 | 3.06 | 3.47 | 3.36 | 4.52 | 3.65 | 3.61 |
| 5000 | **3.07** | **3.48** | **3.41** | **4.53** | 3.65 | **3.63** |
| 10000 | 3.02 | 3.45 | 3.35 | 4.49 | 3.65 | 3.59 |

### A.3.2 OBJECTIVE AS LOSS FUNCTION VS. LOSSAGENT

In the loss repository of LossAgent, we adopt commonly used reference-based loss functions for different tasks, such as L1 loss, LPIPS loss, and perceptual loss. LossAgent achieves the various optimization goals by adjusting the weights of these stable loss functions. However, as mentioned in Section 3.1 of the main paper, some advanced no-reference (NR) IQA metrics can be applied as the loss function, such as CLIPIQA (Wang et al., 2023)/NIQE (Mittal et al., 2012). The primary reason for not using these metrics as loss functions in image processing model training is that such NR loss functions may lead to instability during the training process. To further validate this, we choose the CLIPIQA/NIQE as the optimization objective, and compare the performance of LossAgent against directly adding CLIPIQA/NIQE loss into the training process in Table 9.

It is noteworthy that, we combine three loss functions mentioned in the main paper with CLIP-IQA/NIQE loss instead of only utilizing CLIPIQA/NIQE to optimize the model, since CLIP-IQA/NIQE cannot provide direct guidance on the structural restoration on images, leading to training collapse. As demonstrated in Table 9, directly adding these NR objectives as loss functions into the training process results in suboptimal performance. On the contrary, LossAgent achieves these optimization goals by leveraging the combination of existing stable loss functions.

### A.3.3 $M$ IN LOSS REPOSITORY

To simplify the reasoning process of the LLM agent, we adopt the commonly-used three loss functions (*i.e.*, $M = 3$) in existing IR methods, as demonstrated in Section 4.1 of the main paper. However, the design of the loss repository enables LossAgent to support more loss functions. To validate this, we expand the $M$ to five (L1, MSE, Perceptual, GAN, LPIPS) in Table 10, which demonstrates that a

Table 9: Quantitative comparisons between directly adding optimization objective (CLIPIQA/NIQE) into training process and LossAgent. Results are reported on real-world image SR (RISR). Notably, adopting CLIPIQA/NIQE themselves as the only loss function results in training collapse, and we do not report these results in the table.

|  | Add CLIPIQA↑ | LossAgent | Add NIQE↓ | LossAgent |
|---|---|---|---|---|
| OST300 | 0.202 | **0.571** | 6.59 | **3.05** |
| RealSRSet | 0.226 | **0.649** | 7.83 | **4.43** |

Table 10: Quantitative comparisons between different loss repositories for LossAgent on classical image SR. ↑ / ↓ indicates higher/lower is better. The best results are **bolded**.

| Metrics | Methods | Datasets | | | | | Avg. |
|---|---|---|---|---|---|---|---|
| | | Set5 | Set14 | BSD100 | Urban100 | Manga109 | |
| NIQE↓ | LossAgent ($M = 3$) | 4.82 | 3.91 | **3.86** | 3.96 | 3.88 | 4.08 |
| | LossAgent ($M = 5$) | **4.73** | **3.85** | 3.93 | **3.92** | **3.77** | **4.04** |
| MANIQA↑ | LossAgent ($M = 3$) | 0.474 | 0.418 | 0.365 | 0.496 | **0.424** | 0.436 |
| | LossAgent ($M = 5$) | **0.478** | **0.423** | **0.376** | **0.509** | 0.422 | **0.442** |
| CLIPIQA↑ | LossAgent ($M = 3$) | **0.788** | **0.718** | 0.679 | **0.643** | **0.729** | **0.711** |
| | LossAgent ($M = 5$) | 0.768 | 0.711 | **0.684** | 0.634 | 0.723 | 0.704 |
| Q-Align↑ | LossAgent ($M = 3$) | 3.07 | 3.48 | 3.41 | **4.53** | **3.65** | 3.63 |
| | LossAgent ($M = 5$) | **3.14** | **3.52** | **3.48** | 4.52 | 3.64 | **3.66** |

larger loss repository achieves slightly better results. We attribute this to the limited reasoning ability of the current LLM. Nevertheless, the improved results still indicate the potential of scalability of LossAgent.

### A.3.4 TESTING IMAGE SET

As a crucial part of generating feedback from external expert models, the choice of the testing image set $\mathcal{I}$ is important. We observe that using the sampled Set14 (Zeyde et al., 2010) as the testing image set achieves a better CLIPIQA score compared to using the sampled DIV2K (Agustsson & Timofte, 2017). We attribute this to the relatively high resolution of the DIV2K images. Since some advanced IQA metrics leverage a pre-trained vision encoder to resize input images, this results in originally similar high-resolution images becoming even harder to distinguish after resizing. Consequently, the IQA model may assign similar or even identical scores to these images, failing to provide useful information to our LossAgent. This can cause the LLM to hallucinate and make unreasonable inferences, leading to incorrect adjustment of loss weights. As a result, we choose Set14 as the testing image set for the classical image SR task. We apply the same principle to the other two tasks.

Table 11: Quantitative comparisons between different iterations for each stage. Results are reported on classical image SR task using Q-Align score. The best results are **bolded**.

| Image Set | Datasets | | | | | Avg. |
|---|---|---|---|---|---|---|
| | Set5 | Set14 | BSD100 | Urban100 | Manga109 | |
| Set14 | **0.788** | **0.718** | **0.679** | **0.643** | **0.729** | **0.711** |
| DIV2K | 0.783 | 0.706 | 0.675 | 0.638 | 0.721 | 0.704 |

### A.3.5 THE ILLUSTRATION OF LOSS WEIGHT CURVES

To provide a more intuitive understanding of how LossAgent updates the loss weights, we provide a visualization of the loss weight curves on the classical image super-resolution (CISR) task in Figure 5. Notably, since the four NR objectives in the Figure are all designed for human perception, the curves

Table 12: Training time analysis of LossAgent on three low-level image processing tasks. "Model Training" denotes the necessary training time required by image processing models, which also serves as the reference time for LossAgent comparisons.

| Task | Each Stage | | | Total |
|---|---|---|---|---|
| | Model Training | LLM Reasoning | Feedback | LossAgent |
| CISR | 566min | 58min | 3min | 627min ↑10.78% |
| RISR | 1176min | 192min | 20min | 1388min ↑18.03% |
| AIR | 516min | 83min | 5min | 604min ↑17.05% |

Table 13: Quantitative comparisons between LossAgent and other methods on classical image SR based on **HAT-S** (Chen et al., 2023a) backbone. ↑ / ↓ indicates higher/lower is better. The best results are **bolded**.

| Metrics | Methods | Datasets | | | | | Avg. |
|---|---|---|---|---|---|---|---|
| | | Set5 | Set14 | BSD100 | Urban100 | Manga109 | |
| NIQE↓ | Pre-trained | 7.06 | 6.29 | 6.05 | 5.45 | 5.21 | 6.01 |
| | Fixed | 4.90 | **4.10** | **3.82** | 4.16 | 4.09 | 4.21 |
| | LossAgent | **4.42** | 4.18 | 3.91 | **4.06** | **3.89** | **4.09** |
| | GT (Ref.) | 5.15 | 4.86 | 3.19 | 4.02 | 3.53 | 4.15 |
| MANIQA↑ | Pre-trained | 0.447 | **0.405** | 0.347 | 0.480 | **0.440** | 0.424 |
| | Fixed | 0.438 | 0.387 | 0.360 | 0.476 | 0.403 | 0.412 |
| | LossAgent | **0.460** | 0.404 | **0.374** | **0.505** | 0.424 | **0.434** |
| | GT (Ref.) | 0.534 | 0.449 | 0.523 | 0.552 | 0.420 | 0.496 |
| CLIPIQA↑ | Pre-trained | 0.620 | 0.512 | 0.536 | 0.500 | 0.637 | 0.561 |
| | Fixed | **0.765** | 0.674 | 0.679 | 0.613 | 0.706 | 0.688 |
| | LossAgent | 0.740 | **0.711** | **0.681** | **0.636** | **0.728** | **0.699** |
| | GT (Ref.) | 0.807 | 0.740 | 0.756 | 0.675 | 0.700 | 0.736 |
| Q-Align↑ | Pre-trained | **3.03** | 3.29 | 2.95 | 4.37 | 3.65 | 3.46 |
| | Fixed | 2.95 | 3.38 | 3.40 | 4.52 | **3.66** | 3.58 |
| | LossAgent | 2.98 | **3.49** | **3.44** | **4.55** | 3.65 | **3.62** |
| | GT (Ref.) | 3.36 | 3.63 | 4.04 | 4.53 | 3.60 | 3.83 |

theoretically exhibit similar trends at a coarse granularity, while their fine-grained preferences differ. Consequently, our LossAgent adjusts loss weights following the theoretical rules while adapting to different optimization objectives.

### A.3.6 FLEXIBILITY OF LOSSAGENT ACROSS DIFFERENT IMAGE PROCESSING MODELS

In this section, we demonstrate the flexibility of LossAgent by applying two recent image restoration backbones as the image processing model. As demonstrated in Table 13 and 14, HAT-S and MambaIRv2-light optimized by LossAgent outperform other methods on almost all benchmarks across four metrics. This further demonstrates the flexibility of LossAgent toward different backbones of image processing models, indicating the great potential of LossAgent.

### A.3.7 TIME ANALYSIS

In this section, we provide the training time analysis of LossAgent in Table 12. As demonstrated, LossAgent introduces approximately 15% extra training time averaged on three training tasks. Compared to the performance improvement brought by LossAgent, we consider the additional training time to be acceptable. Notably, there is no additional time overhead during testing since we only introduce LossAgent during training.

Table 14: Quantitative comparisons between LossAgent and other methods on classical image SR based on **MambaIRv2-light** (Guo et al., 2024b;a) backbone. ↑ / ↓ indicates higher/lower is better. The best results are **bolded**.

| Metrics | Methods | Datasets | | | | | Avg. |
|---------|---------|------|-------|--------|---------|----------|------|
| | | Set5 | Set14 | BSD100 | Urban100 | Manga109 | |
| NIQE↓ | Pre-trained | 6.78 | 6.22 | 6.21 | 5.49 | 5.14 | 5.97 |
| | Fixed | 4.95 | 4.10 | 3.85 | 4.08 | 3.84 | 4.17 |
| | LossAgent | **4.42** | **4.03** | **3.78** | **4.04** | **3.72** | **4.00** |
| | GT (Ref.) | 5.15 | 4.86 | 3.19 | 4.02 | 3.53 | 4.15 |
| MANIQA↑ | Pre-trained | 0.428 | 0.379 | 0.324 | 0.454 | 0.432 | 0.403 |
| | Fixed | 0.415 | 0.381 | 0.333 | **0.470** | 0.410 | 0.402 |
| | LossAgent | **0.435** | **0.395** | **0.348** | 0.464 | **0.436** | **0.415** |
| | GT (Ref.) | 0.534 | 0.449 | 0.523 | 0.552 | 0.420 | 0.496 |
| CLIPIQA↑ | Pre-trained | 0.601 | 0.498 | 0.504 | 0.484 | 0.620 | 0.541 |
| | Fixed | 0.749 | **0.689** | 0.668 | 0.619 | 0.711 | 0.687 |
| | LossAgent | **0.756** | 0.680 | **0.673** | **0.623** | **0.726** | **0.692** |
| | GT (Ref.) | 0.807 | 0.740 | 0.756 | 0.675 | 0.700 | 0.736 |
| Q-Align↑ | Pre-trained | 2.92 | 3.18 | 2.80 | 4.29 | 3.63 | 3.37 |
| | Fixed | 2.82 | 3.33 | 3.25 | 4.52 | 3.61 | 3.50 |
| | LossAgent | **2.91** | **3.48** | **3.34** | **4.61** | **3.65** | **3.60** |
| | GT (Ref.) | 3.36 | 3.63 | 4.04 | 4.53 | 3.60 | 3.83 |

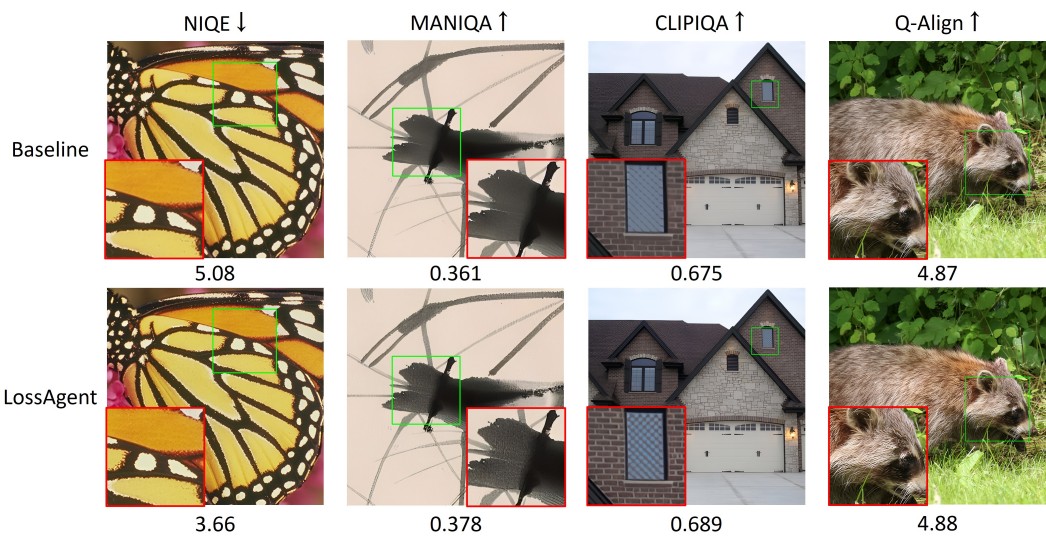

Figure 4: Qualitative comparisons between baseline and LossAgent on real-world image super-resolution across four optimization objectives. Zoom in for best views.

### A.3.8 MORE QUALITATIVE RESULTS

We provide additional qualitative comparisons between LossAgent and other methods on the real-world image super-resolution (RISR) task in Figure 4. As illustrated, the SR model optimized by LossAgent restores more vivid textures that align with optimization objectives.

### A.4 CASE STUDY

In this section, we provide a case study on classical image super-resolution in Figure 6 to help readers better understand the process of LossAgent. As demonstrated, LossAgent is capable of analyzing the relationships between loss weights and score feedback from historical prompt (we mark such analysis

in green). Moreover, LossAgent updates new loss weights considering not only these relationships but also the functionality of each loss function (we mark such thoughts in red). To get the updated loss weights, we use a python program to parse the pattern "L1:Perceptual:GAN=0.7:0.3:0.05" into the numeric array "[0.7, 0.3, 0.05]". Therefore, the correctness of this pattern is important. As analysed in Section 4.3 of the main paper, we use rule-based formatting constraints, which is helpful for LLaMA3 model.

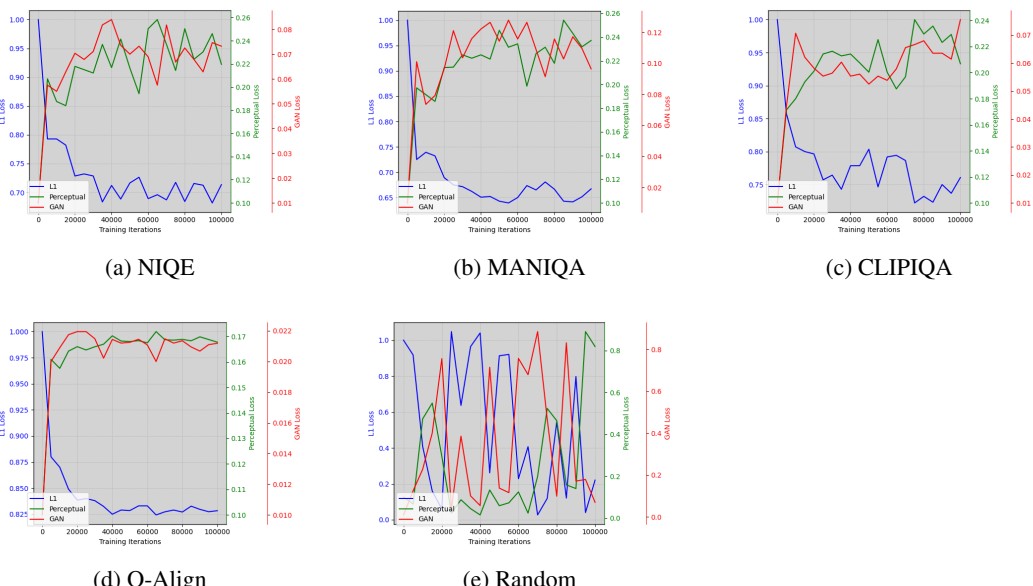

(a) NIQE

(b) MANIQA

(c) CLIPIQA

(d) Q-Align

(e) Random

Figure 5: Illustration of loss weight curves on classical image super-resolution task across four optimization objectives. Zoom in for better views.

**Case Study – NIQE, 25000 iters., Response of one sampled image from Set14**

**System Prompt**

You are a helpful assistant that help people develop loss weights of an SR model. You will receive some historical weights, and score feedback from experts. The SR model is trained with each historical weight for fixed steps of parameter updates sequentially. The historical weights demonstrated the loss weight changes from the beginning of training. The feedback includes the score changes of SR images from the beginning of training in detail. The score depicts the overall quality of images. The lower score stands for better quality. Your ultimate goal is to help the SR model achieve a lower score feedback.

**Historical Prompt**

Feedback: Initial loss weights of L1 loss, Perceptual loss and GAN loss is 1.0:0.1:0.01. The initial score of the image before training is 4.7087. The score of the image at this step is 2.9642.
Step 5000 weights is 0.79286:0.20714:0.0575. The score of the image at this step is 2.5495
Step 10000 weights is 0.79286:0.1875:0.055. The score of the image at this step is 2.7706
Step 15000 weights is 0.78214:0.18393:0.06286. The score of the image at this step is 2.8586
Step 20000 weights is 0.72857:0.21786:0.07036. The score of the image at this step is 2.705

**Customized Needs Prompt**

Based on your observation of all historical loss weights, the scores and your knowledge about these losses, please develop a new loss weights to train the SR model.
You should follow these rules:
1. The output should in format 'Weights: L1:Perceptual:GAN=?:?:? (for example: L1:Perceptual:GAN=1.0:0.1:0.01)' 'Thoughts: (The reason you give this weights)'.
2. Your ultimate goal is to help SR model achieve a lower score feedback (which indicates a better image quality).
3. You can randomly increase or decrease the weight of each loss by a small amount if historical weights are not enough for making a decision.

**LLM's Feedback**

Based on the historical data and the goal of improving image quality (lower score), here's a proposed set of updated loss weights for your SR model:
Weights: L1:Perceptual:GAN=0.7:0.3:0.05
Thoughts: Decreasing the L1 loss weight and slightly increasing the Perceptual loss weight seems beneficial based on the observed score changes. Initially, the model started with a high L1 loss weight, which has been gradually reduced over training steps. This reduction correlated with improvements in image quality scores. The slight increase in Perceptual loss weight helps in preserving more perceptual details, further enhancing image quality. GAN loss weight is kept relatively low to prevent overfitting to the discriminator and maintain visual fidelity.
This adjustment aims to leverage the learning from historical weight changes, emphasizing a balance that has shown incremental improvements in image quality scores over training steps. Adjusting weights further could be iterative, guided by continued monitoring of image quality feedback during training.

Figure 6: A case study of LossAgent on classical image super-resolution task at 25000 iterations. The optimization objective is NIQE.

