# OpenReview forum: "LossAgent: Towards Any Optimization Objectives for Image Processing with LLM Agents"
_ICLR.cc/2026/Conference — ICLR 2026 Conference Withdrawn Submission_

### Official Review · Reviewer_4JLs · 2025-10-28

**Soundness:** 3
**Presentation:** 3
**Contribution:** 3
**Rating:** 6
**Confidence:** 3

**Summary:**

The paper proposes LossAgent, a framework that leverages a large language model (LLM) to dynamically design and adjust loss functions during image restoration training. The LLM analyzes task descriptions and feedback (including non-differentiable metrics like NIQE or CLIPIQA) to generate adaptive combinations of common loss terms (L1, perceptual, LPIPS). Experiments show improved perceptual quality over fixed or random loss weighting baselines.

**Strengths:**

1. The conceptual contribution is strong — using LLMs to reason about optimization objectives is fresh.
2. The framework is well-structured and interpretable, with a clear prompting process that allows for transparent reasoning.
3. Experiments, while limited in scope, consistently show perceptual improvements and visually convincing outputs.

**Weaknesses:**

1. The implementation remains quite constrained — only three basic loss terms (L1, perceptual, and LPIPS) are employed. This narrow setup makes the paper’s claim of “towards any objective” feel overstated.

2. The paper provides no analysis of training stability or convergence, despite the fact that the optimization objective changes dynamically throughout training.

3. While Table 12 reports the training time overhead, the paper does not discuss GPU memory consumption or hardware requirements. I am curious about the actual computational overhead introduced by incorporating the LLM into the training loop.

**Questions:**

Expand the loss library beyond the three basic terms (L1, perceptual, LPIPS) and evaluate the framework on a wider variety of objectives. Alternatively, provide a clear justification for why these three losses are sufficient to represent the range of “any optimization objective” claimed in the paper.

---

### Official Review · Reviewer_62YK · 2025-10-31

**Soundness:** 3
**Presentation:** 3
**Contribution:** 2
**Rating:** 2
**Confidence:** 4

**Summary:**

The authors propose to utilize large language models (LLMs) as loss agents, to aid the design of optimization objectives via dynamic adjustment of the weighted combination of losses from a repository of differentiable loss functions. Specifically, the authors target non-differentiable perceptual image quality assessment (IQA) metrics such as NIQE and CLIPIQA, and the proposed agent works by generating new loss weights for the subset of differentiable loss functions. Experimental results show the proposed method being applied across several image processing models (SwinIR, PromptIR) and well-known datasets, showing consistent improvements over fixed or random loss weight baselines.

**Strengths:**

- The proposed formulation of redefining the optimization process through LLM-based agent is well-motivated, and seems to be generalizable to other applications with non-differentiable objective functions.

- The objective of the proposed method is explained in a straightforward manner, with simple formulation which can be easily implemented and reproduced by other researchers.

- The authors performed extensive experimental validations on multiple well-known image restoration baselines, and they also conducted some ablation experiments to facilitate further understanding for the proposed work.

**Weaknesses:**

- There were previous works with similar motivation, which is using LLMs as agents to aid the optimization process, hyperparameter selection, and architecture design [a,b,c,d,e]. Although overlap in motivation widely vary across publications, these works are worth mentioning at the related works section.
  - [a] AgentHPO: Large Language Model Agent for Hyper-Parameter Optimization, Liu et al., CPAL 2025.
  - [b] Language Models as Black-Box Optimizers for Vision-Language Models, Liu et al., CVPR 2024.
  - [c] On the Convergence of Large Language Model Optimizer for Black-Box Network Management, Lee t al., IEEE Trans. Comm. 2025.
  - [d] Large Language Model Enhanced Particle Swarm Optimization for Hyperparameter Tuning for Deep Learning Models, Hameed et al., IEEE Open Journal of the Computer Society 2025.
  - [e] InstructZero: Efficient Instruction Optimization for Black-Box Large Language Models, Chen et al., ICML 2024.

- Related to the aforementioned weakness, compared to previous works with similar motivations, the formulation of the proposed work seems too simple, with LLM only predicting a set of weights that control the amount of contribution of each loss term without any high-level understanding of the IQA metric and role of each loss terms. Also, it heavily relies on the user input prompt, where it should also be learned through optimization.

- Comparison with baseline methods of "fixed" and "random" settings seem insufficient and weak, with possible strong baselines that can be used for optimizing non-differentiable objective functions, such as reinforcement learning (simple REINFORCE algorithm should suffice) and sampling-based methods (simple particle filter or MCMC should suffice).

- Although the proposed scheme shows empirical performance gains, there are no theoretical grounds on the algorithm's convergence, and this seems to be the case for Fig. 5 in the supplementary material. How sensitive is the proposed framework under different iteration numbers? Is the optimization iteration picked by choosing the best-performing model?

**Questions:**

Please refer to the aforementioned weaknesses. The paper brings some interesting ingredients to show, and the proposed scheme is simple and straightforward. However, limitations in the contributions and lack of theoretical analysis should be addressed.

---

### Official Review · Reviewer_CFTc · 2025-11-01

**Soundness:** 3
**Presentation:** 3
**Contribution:** 3
**Rating:** 2
**Confidence:** 4

**Summary:**

The paper introduces a new learning strategy for image restoration tasks. In the process, the paper proposes to employ an LLM as loss agent that is prompted to find loss coefficients. The paper carefully designs a prompt and context consisting of system prompts, feedback (historical), and customized prompts, which are used by an agent to reason about given feedback and produce loss weights during training. The experimental results demonstrate the effectiveness of dynamically adjusting loss coefficients via agent.

**Strengths:**

- The proposed idea of employing an LLM agent to incorporate feedback from non-differentiable metric to dynamically adjust loss coefficients of differentiable loss functions is interesting.
- The ablation study experiments are comprehensive.
- Despite the use of LLM, the training overhead is not as large as one would expect.

**Weaknesses:**

- The paper claims "achieving the required optimization trajectory for any customized optimization objectives". However, this seems to be overstated claim, since the proposed method takes the feedback from the non-differentiable metric of interest and reweighs a different differentiable loss function.
- The motivation of the paper is closely related to a line of works on meta-learning that aims to automate the search of hyperparameters, in particular, the work on meta-learned loss functions, meta-learning with rewards, and works that use LLMs for hyperparameter tuning [A-J]. Yet, the paper does not provide discussions and experimental comparisons against related works.
- The paper, instead, shows comparisons against naive approaches, such as random or fixed coefficients.
- The performance improvement is marginal, especially on MANIQA and Q-Align.
- The proposed method requires pre-trained checkpoints, which makes the statement that the proposed method performs end-to-end optimization a bit overstated.


[A] Maclaurin et al., Gradient-Based Hyperparameter Optimization Through Reversible Learning. ICML 2015.
[B] Franceschi et al., Bilevel Programming for Hyperparameter Optimization and Meta-Learning. ICML 2018.
[C] Li et al., AutoLoss-Zero: Searching Loss Functions from Scratch for Generic Tasks. CVPR 2022.
[D] Baik et al., Meta-Learning with Task-Adaptive Loss Functions for Few-Shot Learning. ICCV 2021.
[E] Bechtle et al., Meta-Learning via Learned Loss. arXiv 2019.
[F] Liu et al., Large Language Model Agent for Hyper-Parameter Optimization. arXiv 2024.
[G] Liu et al., Large Language Models to Enhance Bayesian Optimization. ICLR 2024.
[H] Xu et al., AutoLoss: Learning Discrete Schedules for Alternate Optimization. arXiv 2018.
[I] Falkner et al., BOHB: Robust and Efficient Hyperparameter Optimization at Scale. ICML 2018.
[J] Joph and Le. Neural Architecture Search with Reinforcement Learning. ICLR 2017.

**Questions:**

- How does the proposed method compare against the works that perform hyperparameter optimization (loss coefficients) with reinforcement learning and/or evolutionary algorithms (with non-differentiable metric as rewards)?
- How does the performance curve on each non-differentiable metric look as the training proceeds for the proposed method, random, fixed coefficients, and works that hyperparameter optimization (loss coefficients) with reinforcement learning and/or evolutionary algorithms (with non-differentiable metric as rewards)?

---

### Official Review · Reviewer_n9Vk · 2025-11-01

**Soundness:** 3
**Presentation:** 3
**Contribution:** 2
**Rating:** 6
**Confidence:** 3

**Summary:**

The paper proposes LossAgent, an LLM-driven controller that adaptively sets the weights of a repository of differentiable losses during training of low-level image processing models. The agent receives external, potentially non-differentiable feedback, plus historical trajectories of loss weights and scores, and outputs new loss-weight combinations via prompted reasoning. This is positioned as a way to optimize towards any objectives while keeping back-prop differentiable through the standard losses.

**Strengths:**

1. Converting external IQA or textual comments into loss weights is pragmatic and avoids unstable NR-loss training; the paper even shows direct NR-loss training can be worse.

2. Consistent SR gains under multiple metrics and also under dual-objective (Q-Align+PSNR), indicating the agent can balance conflicts.

3. Prompt ablations (system vs. historical vs. format rules) support the design choices; helpful for reproducibility and future extensions.

4. The paper fairly notes weaker gains for all-in-one restoration and attributes it to smaller inter-stage visual differences limiting informative feedback. This limitation is important for scope.

**Weaknesses:**

1. The underlying optimization remains a weighted sum of standard losses (no new loss, model, or training objective), and the agent’s novelty lies mainly in the control policy selection by LLM prompts. The technical depth is lighter than methods that derive new differentiable surrogates or link gradients to feedback.

2. Any objective implicitly assumes a good differentiable proxy basis exists in the repository. The paper shows CLIPIQA/NIQE-as-loss can be unstable and that LossAgent does better, but it remains unclear how far this extends beyond the chosen objects (L1 / perceptual / GAN / LPIPS).

3. Stability and reproducibility of the agent loop (variance across runs/LLM sampling; sensitivity to stage length M and repository size) are only partially probed; end-to-end variance or sensitivity to LLM temperature/seed is not reported.

4. What is the additional cost of each LLM inference in each stage and its percentage of the total training time? Is it more efficient than "grid/Bayesian hyperparameter tuning"?

**Questions:**

Please refer to the weaknesses.

---

### Note · Authors · 2025-11-13

**Comment:**

We thank the great efforts made by reviewers and ACs.

**Withdrawal Confirmation:**

I have read and agree with the venue's withdrawal policy on behalf of myself and my co-authors.